# Dynamic Structure Factors in Two Dimensional $Z_2$ Lattice Gauge Theory

**Fırat Yılmaz[1,2], Erich J Mueller [2⋆]**

**1** ENQUBT, Integrated Quantum Information Technologies A.Ş., İstanbul 34467, Turkey
**2** Laboratory of Atomic and Solid State Physics, Cornell University, Ithaca, New York

⋆ em256@cornell.edu

## Abstract

We numerically calculate the dynamic structure factor of the simplest two dimensional $Z_2$ lattice gauge theory. This provides an important benchmark for future experiments which will explore the dynamics of such models. As would be expected, the spectrum is gapped away from the critical point, and can be understood in terms of the elementary excitations.

## 1   Introduction

Lattice gauge theories have become an important target for quantum simulation [1–4]. Such simulations would be used for two parallel and interrelated research goals: Modeling the central phenomena in particle physics and exploring the emergent physics of constrained quantum systems [5, 6], Due to the challenges of numerically studying time dependent quantum mechanical many-body problems, analog simulations are particularly appealing for the insight

they give into dynamics. We can use quantum Monte Carlo of lattice QCD to calculate Hadron masses [7], but we cannot use quantum Monte Carlo to directly reproduce a collision experiment. To provide a benchmark for these experiments, we numerically study the dynamic structure factor of the simplest pure $Z_2$ lattice gauge theory in two dimensions.

In a Lagrangian framework, lattice gauge theories are formulated as models where "matter" degrees of freedom sit on space-time lattice sites, while "gauge" degrees of freedom sit on links [8]. The link variables act as *connections*, relating the local Hilbert space at separate lattice points. This description of the system has a built-in redundancy, as the action is chosen to be invariant under simultaneously applying a rotation to an individual site, and the links connected to it. These rotations form a group, in our case $Z_2$. We will work with at *pure* gauge theory, where there are no matter particles. This is analogous to studying electromagnetism in the absence of charges, and we are effectively studying the $Z_2$ analog of light. Unlike electromagnetism, our gauge excitations are strongly interacting, and have a gapped spectrum.

One dimensional lattice gauge theories have been studied using superconducting circuits [9–16] and cold atoms [17–25]. A number of proposal have emerged for scaling these up to higher dimension [26]. There are exciting ideas about implementing Lorentz symmetry [27], and producing analogs of quantum electrodynamics or quantum chromodynamics [28]. These experiments are typically well suited to exploring dynamics, such as the ones studied here.

Quantum simulations are generally divided into two classes: Digital approaches use a a sequence of gates to mimic the Trotterized time evolution [2, 29–32], while analog approaches directly implement the desired Hamiltonian [33, 34]. The gauge constraints can be implemented by carefully fine-tuning the parameters, or the geometry can be chosen so that they are automatically obeyed. For example, the constrained dynamics in many Rydberg atom experiments [20, 21, 35, 36] are equivalent to lattice gauge theories [37–40]. Tilted lattices lead to similar constraints [41, 42]. Additionally, one can implement models with emergent low energy gauge theory descriptions [43, 44]. There are also a rich set of experiments where density dependent hoppings produce Hamiltonians which have some resemblance to gauge theories, even if they may not display the full gauge symmetry [45–51].

The simplest dynamics experiments involve linear response: One excites the system with a weak probe, and then monitors how some physical property evolves [52]. Equivalent information is given by probes such as electron or neutron scattering. By the fluctuation-dissipation theorem, these response functions are encoded in dynamical correlation functions. In a translationally invariant setting these correlation functions can be decomposed into momentum channels and are described as *dynamic structure factors*. We use *exact diagonalization* to calculate the dynamic structure factors of a $Z_2$ lattice gauge theory on small systems of up to 36 bonds.

In Sec. 2 and 3 we describe our model and numerical technique. In Sec. 4 we present our numerical results, and connect them to known properties of the model. Section 5 contains conclusions and gives an experimental outlook.

# 2 Model

We consider the $Z_2$ lattice gauge theory introduced by Wegner in 1971 [53]. It is described by a 2D square array, where a 2-level system (quantum spin) sits on each link. The Hamiltonian consists of Pauli $X$ and $Z$ operators. Each plaquette term, $U_p = Z_i Z_j Z_k Z_l$, involves a product of Pauli $Z$ operators, with $i, j, k, l$ labeling the four spins making up the square plaquette $p$. It also contains single site Pauli $X$ terms, and we will write

$$H = -\cos\theta \sum_p U_p - \sin\theta \sum_i X_i. \tag{1}$$

⁶³ This Hamiltonian commutes with all of the star operators $W_s = X_i X_j X_k X_l$, where $i, j, k, l$ label
⁶⁴ the four spins contained in the star $s$, ie. the four spins on the bonds connected to a single
⁶⁵ site – see Fig. 1. This can be interpreted as a $Z_2$ gauge theory in the temporal gauge, where
⁶⁶ one takes the time-like component of the vector potential to vanish. The star operators are the
⁶⁷ generators of spatial gauge transformations, and the physical (gauge invariant) states are the
⁶⁸ eigenstates of $W_s$ with eigenvalue 1. The operators $X_i$ play the role of an electric field, while
⁶⁹ $U_p$ can be thought of as a magnetic flux.
⁷⁰     As a function of $\theta \in [0, \pi/2]$, the ground state of Eq. (1) undergoes a phase transition at
⁷¹ $\theta_c = 0.104\pi$ [8, 54]. For $\theta < \theta_c$ the system is a $Z_2$ spin liquid, with a 4-fold degeneracy on
⁷² the torus. This is referred to as a deconfined phase, as the energy associated with a pair of
⁷³ electric defects (ie. locations where $W_s = -1$) is independent of their separation. For $\theta > \theta_c$
⁷⁴ one has a paramagnet. The ground state is non-degenerate on the torus, and electric defects
⁷⁵ are confined: The energy of a pair of electric defects varies linearly with their separation.
⁷⁶     For $\theta = 0$ the ground states corresponds to that of the Toric Code [55], they are the
⁷⁷ simultaneous eigenstate of the $U_p$ and $W_s$, all with eigenvalue +1. The excited states of excess
⁷⁸ energy $2n$ instead have $n$ plaquettes for which $U_p |\psi\rangle = -|\psi\rangle$. These are highly degenerate, as
⁷⁹ there are $\binom{N_p}{n}$ ways of distributing the flipped plaquettes, where $N_p$ is the number of plaquettes
⁸⁰ in the system. For small non-zero $\theta$, this degeneracy is broken, giving a bandwidth $\sim \sin(\theta)$.
⁸¹ Periodic boundary conditions force $n$ to be even.
⁸²     For $\theta = \pi/2$ the ground state is a product state, where every spin is a +1 eigenstate of
⁸³ the $X$ operator, denoted $|\rightarrow\rangle$. Excited states are again degenerate: If $m$ spins are flipped,
⁸⁴ the excess energy is $2m$. The gauge constraint $W_s = 1$ restricts the allowed configurations
⁸⁵ of flipped spins. Any allowed spin configuration can be constructed by flipping spins along a
⁸⁶ sequence of closed paths. This can be accomplished by applying some product of $U_p$ operators,
⁸⁷ supplemented by the non-trivial loops, corresponding to chains of $Z$ operators which trace out
⁸⁸ a non-contractible path. There are two such topologically nontrivial loops on a torus. If we
⁸⁹ restrict ourselves to the space of states that can be reached by applying the $U_p$ to the grounds
⁹⁰ state, then $m$ must be even and the smallest allowed $m$ is 4. If $\theta$ deviates slightly from $\pi/2$,
⁹¹ then the degeneracies are broken, yielding bands of width $\sim \cos(\theta)$.
⁹²     In thinking about the properties of this model, it is sometimes convenient to perform a
⁹³ duality transformation, introducing dual spins at the center of each plaquette, whose Pauli
⁹⁴ operators are $\bar{X}$ and $\bar{Z}$. One envisions a canonical transformation which maps $U_p$ onto $\bar{X}_p$.
⁹⁵ For each operator $X_i$ which sits between plaquettes $p$ and $p'$, we identify an operator on the
⁹⁶ dual spins $\bar{Z}_p \bar{Z}_{p'}$. The map from $\{X, Z\} \to \{\bar{X}, \bar{Z}\}$ can readily be seen to be Canonical, as it
⁹⁷ maintains the commutation relations. There are half of many dual spins than gauge spins –
⁹⁸ which is accounted for by the constraints $W_s = 1$. The mapping, however, is non-local, so care
⁹⁹ must be taken about boundary conditions. Nonetheless, the spin-liquid phase maps onto a
¹⁰⁰ paramagnet in the dual basis, where $\bar{X}_p = 1$. The paramagnet of gauge spins maps onto a dual
¹⁰¹ spin ferromagnet. The duality transformation on Eq. (1) maps it onto

$$H \quad = \quad -\cos\theta \sum_i \bar{X}_i - \sin\theta \sum_{\langle ij \rangle} \bar{Z}_i \bar{Z}_j \tag{2}$$

¹⁰² which is the transverse-Field Ising model with $\Gamma = \cos\theta$ and $J = \sin\theta$. According to [56],
¹⁰³ the transition in the 2D transverse field Ising model is at $\Gamma/J = 3.044(2)$, corresponding to
¹⁰⁴ $\theta = 0.104\pi$.

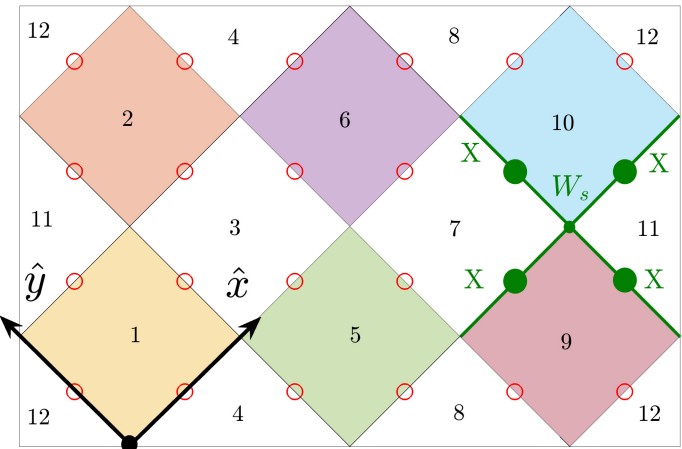

Figure 1: Gauge spins, marked by red circles, sit on the bonds of a square lattice, oriented 45° with respect to the periodic boundary conditions. We color alternating plaquettes, and number them. The gauge spins form a $L_x \times L_y$ square lattice. In this case $L_x = 6$ and $L_y = 4$. The plaquette operator $U_p$ is a product of Pauli $Z$ operators on the four spins on the edges of the plaquette labeled by the integer $p$. The star operators, $W_s$ correspond to a product of Pauli $X$ operators on the four spins surrounding a vertex. On the far right of the figure, one example is highlighted. The $\hat{x}$ and $\hat{y}$ directions are shown, which are used to define the translation operators $T_x$ and $T_y$.

## 3 Methods

To study the dynamical properties of Eq. (1), we consider small systems, where we can fully enumerate a basis for the many-body Hilbert space. We write the Hamiltonian as a matrix in this basis, and then numerically calculate the spectrum. Dynamical correlation functions are found by writing the relevant operators as finite dimensional matrices.

The Hilbert space for an array of 36 spins contains nearly $10^{11}$ states. If these are gauge spins on the edges of a square grid, the gauge constraints reduces this Hilbert space to $1.3 \times 10^5$ states. If one breaks this down by momentum sector, the largest sector has $7.3 \times 10^4$ states in it. In our approach these are the largest matrices that we will need to construct or diagonalize. Unfortunately, due to the exponential scaling with system size, it is very challenging to extend this technique to larger systems. As seen in Fig. 1 we work with periodic arrays of plaquettes, oriented with the lattice oriented at 45° with respect to the boundaries. The gauge spins for a $L_x \times L_y$ square lattice. To construct the Hilbert space we start with a reference state, where every spin is pointing in the $+X$ direction. We then conduct a breadth-first search where we systematically apply products of the plaquette operators $U_p$, adding the new states to our list of Hilbert space states, and building up the matrix elements of $H$, stored as a sparse array. Note, this technique only produces one quarter of the entire Hilbert space: There are four distinct sectors, which are connected by acting with topologically non-trivial chains of $Z$ operators. In the finite size system, the ground state is always in the sector that we explore, and none of our response functions involve operators which take us out of this sector. Thus for this study it would be wasteful to generate the other sectors.

In this basis we construct translation operators $T_x$ and $T_y$, corresponding to shifting the underlying square lattice. For each basis state $|\psi\rangle$, we find the sets of states which are formed by applying powers of $T_x$ and $T_y$. From these we construct momentum eigenstates. Each momentum sector is closed under the action of $H$, and we use a dense matrix eigensolver to separately diagonalize each one. In addition to being numerically efficient, this approach allows us to fix a momentum label to each energy eigenstate. The resulting eigenvalue relation is $H|\vec{k}, \lambda\rangle = E_\lambda(\vec{k})|\vec{k}, \lambda\rangle$, where $\lambda$ is a band index, which labels the different eigenstates. We caution that since we restrict ourselves to one of the four topologically distinct sectors, our spectra will not display the four-fold topological degeneracy that is present in the deconfined phase.

We calculate two dynamical correlation functions

$$S_{XX}(\vec{k}, \omega, \theta) = \langle X_{-\vec{k}}(\omega - H(\theta) + E_0 - i\eta)^{-1} X_{\vec{k}}\rangle_0 \tag{3}$$

$$S_{UU}(\vec{k}, \omega, \theta) = \langle U_{-\vec{k}}(\omega - H(\theta) + E_0 - i\eta)^{-1} U_{\vec{k}}\rangle_0 \tag{4}$$

where $\eta$ is a small positive number which we use to smooth the spectra. Here

$$X_{\vec{k}} = \sum_j X_j e^{i\vec{k}\cdot\vec{r}_j}, \quad U_{\vec{k}} = \sum_p U_p e^{i\vec{k}\cdot\vec{R}_p}, \tag{5}$$

where $\vec{r}_j$ is the location of the $j$'th gauge spin and $\vec{R}_p$ is the center of the plaquette $p$. The operators $X_{\vec{k}}$ and $U_{\vec{k}}$ connect many-body states with momentum $\vec{p}$ and $\vec{p} + \vec{k}$. We only need to construct the matrix elements between the $\vec{k} = 0$ sector and $\vec{k} = \vec{p}$ sector. We then project Eqs (3) and (4) onto the energy eigenstates, reducing it to a series of matrix multiplications. These correlation functions involve gauge invariant quantities. Complementary information can be extracted from spin correlation functions which take one between super-selection sectors [57].

# 4  Results

Figure 2 shows the spectrum $E_\lambda(\vec{k})$, as a function of $\theta$ for $\vec{k} = (0,0)$ and $\vec{k} = (1,1)\pi$ – taking the $6 \times 4$ lattice of gauge spins illustrated in Fig. 1. Only a single topological sector is shown. For this lattice there are a total of 12 distinct $k$ points in the first Brillouin zone, but we only show two of them.

At $\theta = 0$ the energies form degenerate bands with $E = 2n$ for even integer $n$. As discussed in Sec. 2, these bands are associated with states with $n$ flipped plaquettes. Even when we restrict the total momentum of the state there is a degeneracy, as one can change the relative momenta of the defects. As one increases $\theta$ the states disperse and the gap $\Delta(\theta)$ between the ground state and the first excited state initially shrinks. It reaches a minimum near $\theta = 0.11\pi$, before growing to $\Delta(\pi) = 8$. The states at $\theta = \pi$ correspond to individual flipped spins, and for a fixed $\vec{k}$ are non-degenerate.

The minimum in $\Delta$ is indicative of the phase transition which occurs in the thermodynamic limit. In Fig. 3a we explore this feature by plotting the minimal gap as a function of system size for a ladder geometry of $N \times 2$ sites. The gap clearly vanishes in the thermodynamic limit. The phase transition can be characterized by a logarithmically divergent entanglement entropy,

$$\mathcal{S}(\theta) = -\frac{1}{4N} Tr\rho_L \log \rho_L, \tag{6}$$

$$\rho_L = Tr_R |\Psi_0(\theta)\rangle\langle\Psi_0(\theta)|. \tag{7}$$

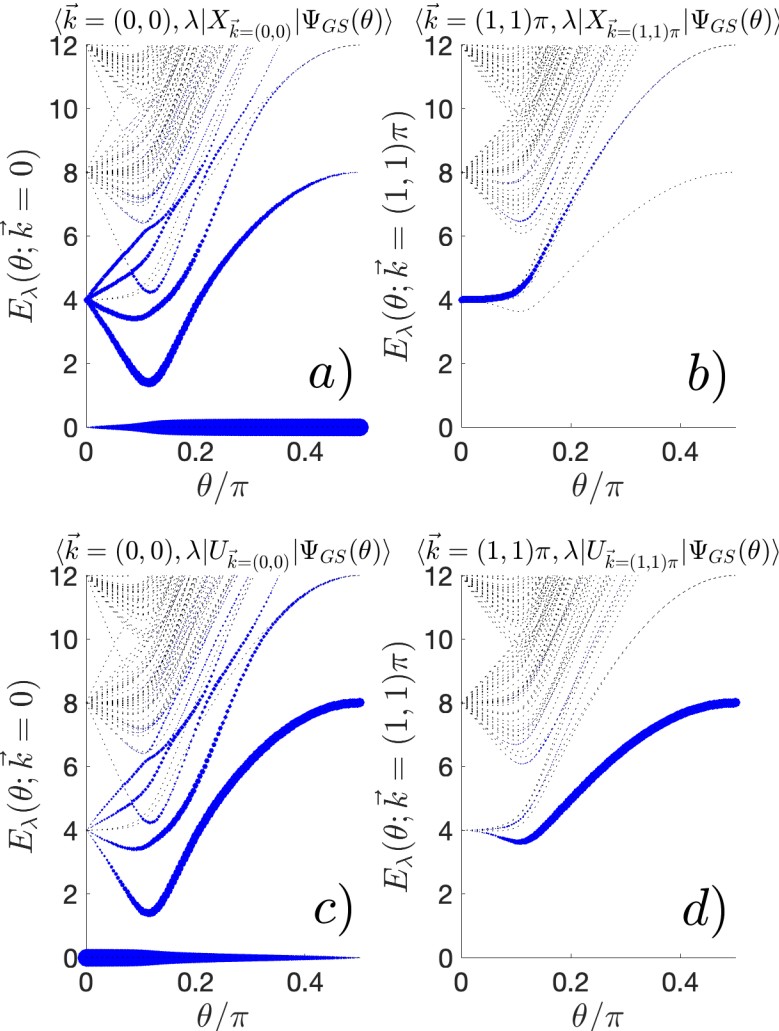

Figure 2: Spectrum of the 6 × 4 lattice of gauge spins illustrated in Fig. 1, in the fundamental topological sector. Left panels show $E_\lambda(\vec{k} = (0,0))$, corresponding to the energies of the states $|\vec{k} = (0,0), \lambda\rangle$ with $\vec{k} = (0,0)$. Here $\lambda$ indexes the states, and we measure energies relative to the ground state. Blue highlighting indicates the overlaps $\langle\vec{k}, \lambda|X_k|\Psi_{GS}\rangle$ (top) or $\langle\vec{k}, \lambda|U_k|\Psi_{GS}\rangle$ (bottom). Thicker lines indicate larger overlaps. The right panels show the same quantities but with $\vec{k} = (1,1)\pi$.

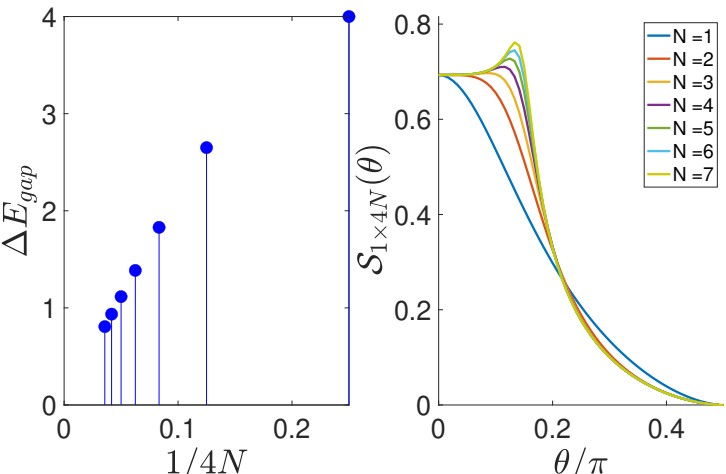

Figure 3: a) The smallest energy gap, $\min_\theta E(\theta) - E_0$ for a $2N \times 2$ spin lattice as a function of $1/N$ where $N \in 1, 2, .., 7$. It vanishes in the thermodynamic limit, indicating that there is a gapless point. b) The entanglement entropy, $\mathcal{S}(\theta)$ as a function of $\theta$, when the $2N \times 2$ lattice is split into two $N \times 2$ regions.

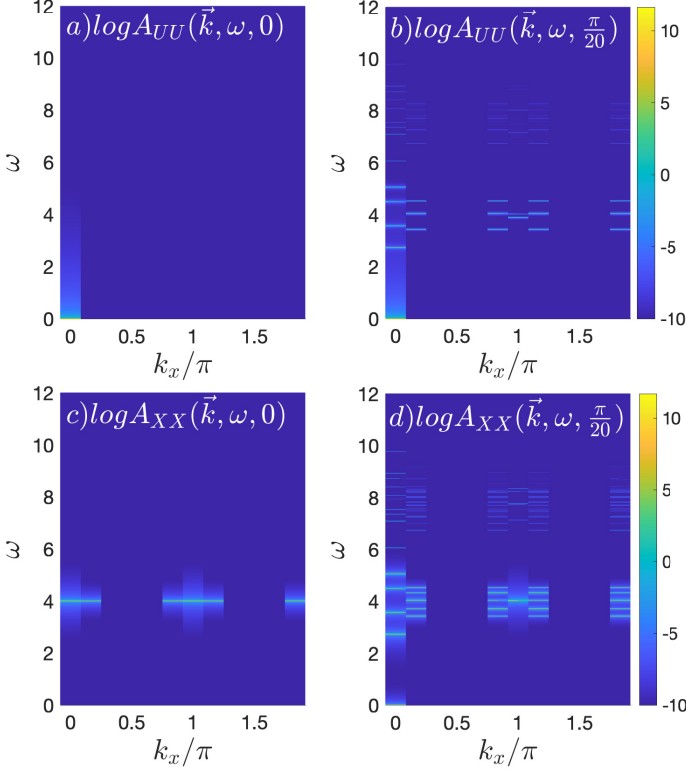

Figure 4: The deconfined phase logarithm of spectral density plots for the dynamical plaquette-plaquette and bond-bond correlation functions, $A_{UU}(\vec{k}, \omega, \theta)$, $A_{XX}(\vec{k}, \omega, \theta)$, respectively. The x-axis corresponds to the x-component of the momentum of the sector: For the 6 geometry shown here, $k_y = k_x + \pi$ is uniquely determined by $k_x$. The y-axis is for the energy, $\omega$.

161       Here $\rho_L$ is the density matrix of the left half of the system, after tracing over the right half.
162 Figure 3b shows the ground state entanglement entropy for ladders of different length. For
163 this geometry at $\theta = 0$ there is one bit of information shared between the two sides of the
164 system, and hence $S(0) = \log(2)$. At $\theta = \pi$ we have a product state and $S(\pi) = 0$. The peak
165 near $\theta = 0.11\pi$ grows with $N$.

166       In addition to showing the many-body spectrum, Fig. 2 shows the matrix elements, $\langle \vec{k}, \lambda | X_{\vec{k}} | \Psi_{\mathrm{GS}} \rangle$
167 and $\langle \vec{k}, \lambda | U_{\vec{k}} | \Psi_{\mathrm{GS}} \rangle$, as blue lines whose thickness is proportional to the matrix elements. The
168 operator $X$ flips a single spin (in the $\hat{z}$ basis). At small $\theta$, this corresponds to flipping two
169 plaquettes, and the bulk of the spectral weight is in the first excited band. For $\theta$ near $\pi/2$ the
170 ground state is an eigenstate of $X$, and the only matrix element is with the ground state. We
171 can make a similar argument with $U$. For small $\theta$ the ground state is an eigenstate of $U$. For $\theta$
172 near $\pi/2$, the $U$ operator flips 4 $x$-spins and the majority of the spectral weight is in the first
173 excited band.

174       Further details about this structure is conveyed by plotting the spectral densities $A_X = \mathrm{Im}\, S_{XX}$
175 and $A_U = \mathrm{Im}\, S_{UU}$, as shown in Figs. 4 and 5. For our $6 \times 4$ lattice, the allowed wave-vectors
176 are $k_x = k_y$ for $k_x = (2n)\pi/3$ and $k_x = k_y \pm \pi$ for $k_x = (2n+1)\pi/6$ where $n = 0, 1, .., 5$. The
177 momentum sectors are uniquely label by only the $k_x$ components and we use this choice. At
178 $\theta = 0$ the $U$ spectral density in Figs. 4a) is proportional to a delta-function at $\vec{k} = (0,0)$ and
179 $\omega = 0$. Increasing $\theta$ to $\pi/20$, in panel b) one sees a sequence of peaks appear near $\omega = 4$.
180 These peaks are nearly dispersionless: The peak locations are identical at all $k_x \neq 0, \pi$. This
181 feature persists to larger $\theta$, as shown in 4 a) and b). At $\theta = \pi/2$ the spectral weight at $\omega = 0$
182 vanishes, and the $U$ spectral density vanishes away from $\omega = 8$.

183       As expected from our previous arguments, the $X$ spectral weight at $\theta = 0$ in Fig. 4 c) is
184 found only at $\omega = 4$, and is dispersionless. At $\theta = \pi/20$, in panel d), the peaks split, but again
185 the frequencies are identical for all $k_x$ except for $0, \pi$. In Fig. 5 c) we see that the $X$ spectral
186 weight shifts to $\omega = 0, \vec{k} = (0,0)$ as $\theta \to \pi/2$.

## 5   Conclusion and Experimental Outlook

188 Dynamics experiments represent the most important application of lattice gauge theory sim-
189 ulators. Here we calculate the gauge invariant dynamic structure factors associated with the
190 pure $Z_2$ lattice gauge theory. Far from the phase transition between the confined and decon-
191 fined phase we find these response functions can be simply understood in terms of the model's
192 quasiparticles.

193       These correlation functions can be studied in a number of systems, including cold atoms,
194 Rydberg atoms, and superconducting circuits. The most challenging aspect of cold atom im-
195 plementations of the Hamiltonian in Eq. 1 is the ring exchange term $U_p = Z_i Z_j Z_k Z_l$. There
196 are, however, a number of strategies which have been proposed, typically involving high order
197 perturbation theory [58]. Constructing this term is simpler in superconducting circuits [59],
198 where one Trotterizes the dynamics, and directly implements gates of the form $e^{-i(Z_i Z_j Z_k Z_l)\delta t}$.
199 As illustrated in Fig. 6, this gate can be implemented by introducing a single ancilliary qubit
200 for each plaquette. One initializes the auxilliary qubit 0 in state 0. One then applies the gates
201 $R = CX_{i0} CX_{j0} CX_{k0} CX_{l0}$, where $CX_{ij}$ is a control-X operation which flips qubit $j$ if and only if $i$
202 is in the excited state. The operation $U$ entangles the ancilla with the plaquette. A phase gate
203 is applied to the ancilla, $e^{iZ_0\phi}$. One once again applies $U$. Straightforward arithmetic gives
204 $R e^{iZ_0\phi} R = e^{iZ_0 Z_1 Z_2 Z_3 Z_4 \phi}$, which is the desired evolution operator when acting on an eigenstate
205 of $Z_0$.

206       The response functions are most easily measured in the temporal domain. For example,
207 at time $t = 0$ one applies a $U$ or $X$ gate. At a later time $t$ one measures either $U$ or $X$. The

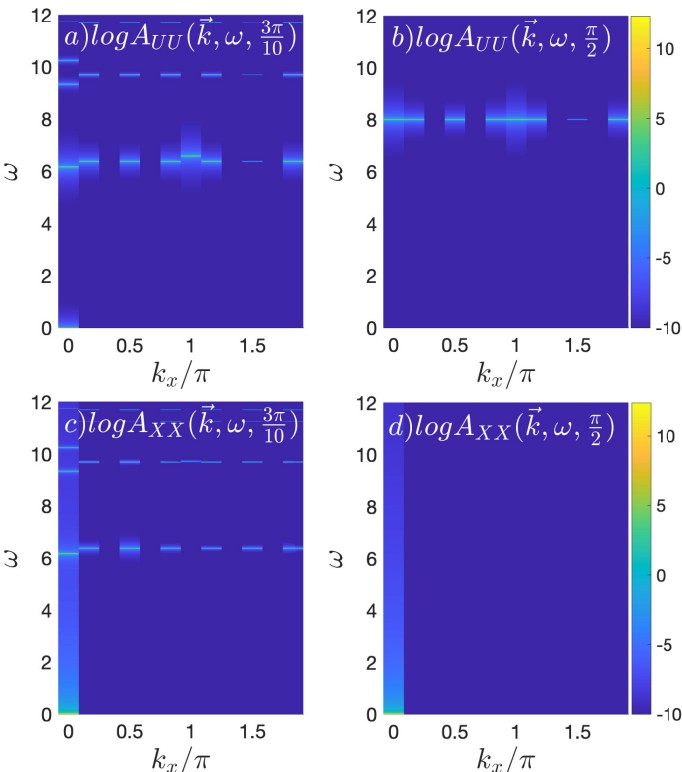

Figure 5: The confined phase logarithm of spectral density plots for the dynamical plaquette-plaquette and bond-bond correlation functions, $A_{UU}(\vec{k}, \omega, \theta)$, $A_{XX}(\vec{k}, \omega, \theta)$, respectively. The x-axis is allocated for lattice momentum sectors (plotted only for $k_x$, yet all of $k_x$ values are unique for our finite size system) and the y-axis is for the energy, $\omega$.

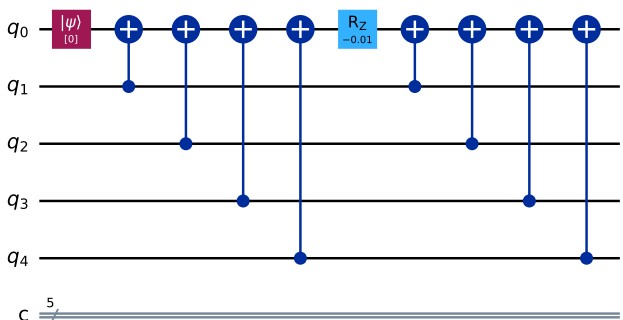

Figure 6: The Trotterized circuit diagram of the plaquette operator for $e^{-i(Z_1 Z_2 Z_3 Z_4)\delta t}$ where $\delta t$ is set to an small number, 0.01. Initially, the ancilla qubit, 0, is set to state 0. One then applies $R = CX_{i0} CX_{j0} CX_{k0} CX_{l0}$ where $CX_{ij}$ is a control-X gate which flips qıbit j if and only if i is in the state 1. One then applies a phase gate to the ancilla qubit $e^{-iZ_0 \delta t}$, shown as a light blue box. The desired Trotterized operator is generated by applying $R$ once again, $Re^{-iZ_0 \delta t} R = e^{-i(Z_1 Z_2 Z_3 Z_4)\delta t}$.

expectation value of that measurement is exactly $S_{XX}(t)$ or $S_{UU}(t)$.

# 6 Funding information

F.Y. is supported by the Deutsche Forschungsgemeinschaft (DFG, German Research Foundation) through project TRR360 A5 and Türkiye Bilimsel ve Teknolojik Araştırma Kurumu (TÜBİTAK) call no.2219. EJM acknowledges support from NSF PHY-2409403.

Authors are required to provide funding information, including relevant agencies and grant numbers with linked author's initials. Correctly-provided data will be linked to funders listed in the Fundref registry.

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
