# Peer review of "Dynamic Structure Factors in Two Dimensional Z2 Lattice Gauge Theory"

_SciPost Physics_

## Round 1 · Referee Report · Anonymous (Referee 1) · 2025-2-16

Strengths

  1. Well-written and pedagogical background description of the model being investigated.
  2. Clear description of the methods, and the results obtained from the method.

Weaknesses

  1. The results displayed in the different figures are not properly interpreted in the text.
  2. The ending of the paper is somewhat abrupt in the sense that the authors introduce the quantum circuit, but immediately conclude, not giving an idea what could be done.

Report

Warnings issued while processing user-supplied markup:

  • Inconsistency: plain/Markdown and reStructuredText syntaxes are mixed. Markdown will be used.
    Add "#coerce:reST" or "#coerce:plain" as the first line of your text to force reStructuredText or no markup.
    You may also contact the helpdesk if the formatting is incorrect and you are unable to edit your text.

==================================================== Report on Dynamic Structure Factors in Two Dimensional $Z_2$ Lattice Gauge Theory ====================================================

In the paper, the authors undertake a pedagogical study of the dynamic structure factors of Z2 lattice gauge theory on a ladder system, and also for lattices when the ladder is widened to a two-dimensional system. While the study itself is technically sound, and a valuable contribution to the literature, some of the points need to be addressed in a more concrete manner.

Therefore, before recommending publication, I urge the authors to address the following points:

  1. Referencing: In my opinion, adding more references can be useful. However, whatever references are added must be meaningful. Reference 7 in line 19 is supposed to report on the quantum Monte Carlo used in lattice QCD to calculate hardon masses, but this particular paper talks about the special methods used in the context of simulations at finite chemical potential, and not about the hadron masses. Please provide appropriate references.

  2. I like the Section 2 describing the model which gives a concise but complete picture about the phases in the model and also talks about excitations of the model in different limits. The authors also mention the connection with the Ising model obtained via Kramers-Wannier dualization.

    Could the authors comment on the nature of degeneracies (as explained in the gauge theory language) when dualized to the spin model?

  3. I suggest the authors to provide a table to show how the number of basis states increase as a function of lattice size (in a given winding sector), perhaps also by states in the momentum (0,0) sector.

    This sentence makes me curious: "If these are gauge spins on the edges of a square grid, the gauge constraints reduces this Hilbert space to $ 1.3 \times 10^5$ states. If one breaks this down by momentum sector, the largest sector has $7.3 \times 10^4$ states in it."

    It seems like the benefit is only a factor of 2 (I am presuming the largest sector is (px,py)=(0,0)). I would have expected a factor lower by the volume, or at least by the larger linear dimension (factor of 6). Even in Section 4, it is stated that there are 12 distinct k values; so I would have expected the number of states in the momentum sector (0,0) to be lower by a factor more than 2.

  4. Eq.(3-5) can help with some more clarification. In particular, what is the exact value of $\eta$ that is chosen, and how does it affect the smoothing of the specta. For example, a curve showing the spectra before and after the smoothing could be useful. Does one need to remove $\eta$ through a limiting process?

    Moreover, eq(5) uses both $r_j$ and $R_p$, but has the same momentum label $k$. Since $r$ and $R$ are related by shifts of one-half lattice spacings, I would have expected the $k$ that appears in the subscript of $X$ and the one that appears in the subscript of $U$ would be related by half-momenta (i.e., with $\frac{\pi n}{L}$). Can the authors please elaborate on this?

    I think the discussion that the authors make in the last two paragraphs of section 4 reflect this? It might make sense to add this information also after Eq.(5).

  5. In the section on the results, the authors say, "In Fig. 3a we explore this feature by plotting the minimal gap as a function of system size for a ladder geometry of NĂ—2 sites. The gap clearly vanishes in the thermodynamic limit."

    While I appreciate the finite size scaling the authors have put in, it remains incomplete because they do not do any fits, but infer the results emperically from the Fig 3a. Looking at Fig 3a, it is clear that it is a non-linear curve. However, a linear extrapolation using the last three, or the last four points (as expected with the vanishing of a gap for a massless theory) should yield an intercept consistent with zero. This is what naively seems to me. This analysis should be provided.

    Also, I think it is instructive to put the gap corresponding to the lattice sizes 4x4 and 6x4 in this figure. It will not probably lie in the same line, but this would be expected since the aspect ratio becomes important in that case. The latter two points are important especially it gives some idea what the 2d system looks like. A chain of 2N x 2 lattice sites ultimately is a quasi 1d system in the limit of large N.

  6. The authors say, "The phase transition can be characterized by a logarithmically divergent entanglement entropy" However, looking at Fig 3b, the logarithmic divergence is not clear at all. Perhaps the authors should plot the peak of the entanglement entropy as a function of N to make their point.

  7. In the caption of Fig 4, should it be, "For the 6x4 geometry shwon here.."? Fig 4 is referred to in the last two sections on Section 4. The authors refer to the figs in terms of 4a, 4b, etc. Although I can infer what they actually are, the a,b, etc are not labelled in the Fig. The authors should properly label the subfigures of Fig 4.

    There is hardly any reference to Fig 5, expect for the last sentence which contains a reference to Fig 5c. I suggest that the authors put in at least one paragraph describing the results of Fig 5 and also label the sub figures accordingly.

  8. The conclusion of the paper seems a bit abrupt. The authors introduce the quantum circuit for the plaquette and the star operators, but do not perform any calculations with it. The quantum ciruit for the Z2 gauge theory is already known in the literature, and even the authors themselves have provided references about the use of the circuits to perform some computations using classical and even quantum hardware.

    The author state that they would want to use the circuits to compute the dynamic structure factors. Could they actually obtain some results with the quantum circuit to show the efficacy of the approach?

  9. Another question is regarding the choice of the methods: exact diagonalization in this case, and with the authors are able to go to 6x4 lattice size. Although this is very impressive, I am curious as to why the authors did not try to implement a Lanczos-type algorithm, given that they are interested in low-energy spectra? It would perhaps have allowed access to larger system sizes. More so, since only the lowest gaps were extracted. Moreover, if the authors showed some real-time dynamics of plaquette excitations disperse, or scatter, the use of only exact diagonalization methods would have carried more weight.

Recommendation

Ask for minor revision

---

## Round 1 · Referee Report · Anonymous (Referee 2) · 2025-2-25

Strengths

1 - The article investigates an aspect of $\mathbb{Z}_2$ lattice gauge theories which is not often discussed, namely the dynamical structure factors.

2 - Generally clearly written

Weaknesses

1 - The significance of the dynamical structure factors could be stated more clearly

2 - Not clear whether it achieves something new from the technical point of view

3 - Some lack of detail regarding the exact diagonalization implementation and results

Report

The authors show some exact diagonalization results for the computation of dynamical structure factors in a pure $\mathbb{Z}_2$ lattice gauge theory, and provide compelling arguments for why such computation is relevant. However, there are several criticisms that need to be addressed before the paper can be considered for publication.

While the introduction provides a good overview of the properties of this particular model, the significance of the dynamical structure factors is only superficially explained. Since computing these quantities is the main goal of the investigation, I suggest expanding the discussion by including more definitions and explaining their application in more detail .

From the technical point of view, it is not clear whether the results constitute a significant improvement over state-of-the art exact diagonalization algorithms. The procedure explained in the paper, namely the method to generate the sparse Hamiltonian matrix and the use of momentum conservation, is well known. It would be good to address explicitly what the limitations are and state which results require the full spectrum of the system and which ones only rely on the computation of the first few eigenstates. Besides, given the unavoidable limitations in system size, it would be beneficial to clarify how the finite-size effects manifest themselves and how they depend on the different geometries chosen. The authors write that they fully diagonalize systems of up to 36 spins, which would be quite remarkable to the best of my knowledge, but I do not see this result presented anywhere in the paper. For example, in Fig. 2 the lattice is 6x4 and in Fig. 3 is 2Nx2 with N up to 7, for a total of 28 spins. This is certainly a point that needs to be clarified.

Overall, I think that this article contains enough results to deserve publication in Scipost after the main point raised here are addressed.

Requested changes

1 - Expand the discussion of the structure factors either in the introduction or in another section.

2 - Explain more clearly the choice of geometry in the different simulations, and how it helps achieving the best results given the limitations in system size. For example, why is a strip of size 2x2N chosen for the plots in Fig. 3?

3 - Add a discussion of how the finite system size affects the results.

4 - I do not understand why $k_y$ is uniquely determined by $k_x$. I would expect them to be independent and quantized depending on $L_x$ and $L_y$ respectively. I suggest adding a clearer explanation.

5 - State more clearly which of the results were obtained for up to 36 spins. If this involve the diagonalization of the full Hamiltonian, it could be beneficial to explain (possibly in an appendix) how such demanding task is technically achieved exploiting translational symmetry only.

Recommendation

Ask for minor revision

---

## Editorial Decision

awaiting_resubmission